# Improving Generative Adversarial Networks with Denoising Feature Matching

**David Warde-Farley & Yoshua Bengio**⋆
Montreal Institute for Learning Algorithms, ⋆ CIFAR Senior Fellow
Université de Montréal
Montreal, Quebec, Canada
`{david.warde-farley,yoshua.bengio}@umontreal.ca`

## Abstract

We propose an augmented training procedure for generative adversarial networks designed to address shortcomings of the original by directing the generator towards probable configurations of abstract discriminator features. We estimate and track the distribution of these features, as computed from data, with a denoising auto-encoder, and use it to propose high-level targets for the generator. We combine this new loss with the original and evaluate the hybrid criterion on the task of unsupervised image synthesis from datasets comprising a diverse set of visual categories, noting a qualitative and quantitative improvement in the "objectness" of the resulting samples.

## 1 Introduction

Generative adversarial networks (Goodfellow et al., 2014a) (GANs) have become well known for their strength at realistic image synthesis. The objective function for the generative network is an implicit function of a learned discriminator network, estimated in parallel with the generator, which aims to tell apart real data from synthesized. Ideally, the discriminator learns to capture distinguishing features of real data, which the generator learns to imitate, and the process iterates until real data and synthesized data are indistinguishable.

In practice, GANs are well known for being quite challenging to train effectively. The relative model capacities of the generator and discriminator must be carefully balanced in order for the generator to effectively learn. Compounding the problem is the lack of an unambiguous and computable convergence criterion. Nevertheless, particularly when trained on image collections from relatively narrow domains such as bedroom scenes (Yu et al., 2015) and human faces (Liu et al., 2015), GANs have been shown to produce very compelling results.

For diverse image collections comprising a wider variety of the visual world, the results have generally been less impressive. For example, samples from models trained on ImageNet (Russakovsky et al., 2014) roughly match the local and global statistics of natural images but yield few recognizable objects. Recent work (Salimans et al., 2016) has sought to address this problem by training the discriminator in a semi-supervised fashion, granting the discriminator's internal representations knowledge of the class structure of (some fraction of) the training data it is presented. This technique markedly increases sample quality, but is unsatisfying from the perspective of GANs as a tool for unsupervised learning.

We propose to augment the generator's training criterion with a second training objective which guides the generator towards samples more like those in the training set by explicitly modeling the data density in addition to the adversarial discriminator. Rather than deploy a second computationally expensive convolutional network for this task, the additional objective is computed in the space of features learned by the discriminator. In that space, we train a denoising auto-encoder, a family of models which is known to estimate the energy gradient of the data on which it is trained. We evaluate the denoising auto-encoder on samples drawn from the generator, and use the "denoised" features as targets – nearby feature configurations which are more likely than those of the generated sample, according to the distribution estimated by the denoiser.

We show that this yields generators which consistently produce recognizable objects on the CIFAR-10 dataset without the use of label information as in Salimans et al. (2016). The criterion appears to improve stability and possesses a degree of natural robustness to the well known "collapse" pathology. We further investigate the criterion's performance on two larger and more diverse collections of images, and validate our qualitative observations quantitatively with the *Inception score* proposed in Salimans et al. (2016).

## 2 BACKGROUND

### 2.1 GENERATIVE ADVERSARIAL NETWORKS

The generative adversarial networks paradigm (Goodfellow et al., 2014a) estimates generative samplers by means of a training procedure which pits a *generator* $G$ against a *discriminator* $D$. $D$ is trained to tell apart training examples from samples produced by $G$, while $G$ is trained to increase the probability of its samples being incorrectly classified as data. In the original formulation, the training procedure defines a continuous minimax game

$$\arg\min_G \ \arg\max_D \mathbb{E}_{\mathbf{x}\sim\mathcal{D}} \log D(\mathbf{x}) + \mathbb{E}_{\mathbf{z}\sim p(\mathbf{z})} \log\left(1 - D\left(G(\mathbf{z})\right)\right) \qquad (1)$$

where $\mathcal{D}$ is a data distribution on $\mathbb{R}^n$, $D$ is a function that maps $\mathbb{R}^n$ to the unit interval, and $G$ is a function that maps a noise vector $\mathbf{z} \in \mathbb{R}^m$, drawn from a simple distribution $p(\mathbf{z})$, to the ambient space of the training data, $\mathbb{R}^n$. The idealized algorithm can be shown to converge and to minimize the Jensen-Shannon divergence between the data generating distribution and the distribution parameterized by $G$.

Goodfellow et al. (2014a) found that in practice, minimizing (1) with respect to the parameters of $G$ proved difficult, and elected instead to optimize an alternate objective,

$$\arg\max_G \mathbb{E}_{\mathbf{z}\sim p(\mathbf{z})} \log D\left(G(z)\right) \qquad (2)$$

at the same time as $D$ is optimized as above. $\log D(G(\mathbf{z}))$ yields more favourably scaled per-sample gradients for $G$ when $D$ confidently identifies a sample as counterfeit, avoiding the vanishing gradients arising in that case with the $-\log(1 - D(G(\mathbf{z})))$ objective.

Subsequent authors have investigated applications and extensions of GANs; for a review of this body of literature, see Warde-Farley & Goodfellow (2016). Of particular note for our purposes is Radford et al. (2015), who provide a set of general guidelines for the successful training of generative adversarial networks, and Salimans et al. (2016), who build upon these techniques with a number of useful heuristics and explore a variant in which the discriminator $D$ is trained to correctly classify *labeled* training data, resulting in gradients with respect to the discriminator evidently containing a great deal of information relevant to generating "object-like" samples.

### 2.2 CHALLENGES AND LIMITATIONS OF GANS

While Goodfellow et al. (2014a) provides a theoretical basis for the GAN criterion, the theory relies on certain assumptions that are not satisfied in practice. Proofs demonstrate convergence of the GAN criterion in the unconstrained space of arbitrary functions; in practice, finitely parameterized families of functions such as neural networks are employed. As a consequence, the "inner loop" of the idealized algorithm – maximizing (1) with respect to (the parameters of) $D$, is infeasible to perform exactly, and in practice only one or a few gradient steps stand in for this maximization. This results in a *de facto* criterion for $G$ which minimizes a lower bound on the correct objective (Goodfellow, 2014).

A commonly observed failure mode is that of full or partial collapse, where $G$ maps a large fraction of probable regions under $p(\mathbf{z})$ to only a few, low-volume regions of $\mathbb{R}^n$; in the case of images, this manifests as the appearance of many near-duplicate images in independent draws from $G$, as well as a lower diversity of samples and modes than what is observed in the dataset. As $G$ and $D$ are typically trained via mini-batch stochastic gradient descent, several authors have proposed heuristics that penalize such duplication within each mini-batch (Salimans et al., 2016; Zhao et al., 2016).

GANs represent a departure from traditional probabilistic models based on maximum likelihood and its approximations in that they parameterize a sampler directly and lack a closed form for the likelihood. This makes objective, quantitative evaluation difficult. While previous results in the literature have reported approximate likelihoods based on Parzen window estimates, Theis et al. (2015) has convincingly argued that these estimates can be quite misleading for high-dimensional data. In this work, we adopt the *Inception score* proposed by Salimans et al. (2016), which uses a reference Inception convolutional neural network (Szegedy et al., 2015) to compute

$$I(\{\mathbf{x}\}_1^N) = \exp\left(\mathbb{E}\left[D_{KL}(p(y|\mathbf{x})\|p(y))\right]\right) \tag{3}$$

where $p(y|\mathbf{x})$ is provided by the output of the Inception network and $p(y) = \int_{\mathbf{x}} p(\mathbf{x})p(y|\mathbf{x})d\mathbf{x} \approxeq \frac{1}{N}\sum p(y|\mathbf{x}_i)$. Note that this score can be made larger by a low-entropy per-sample posterior (i.e. the Inception network classifies a given sample with greater certainty) as well as a higher entropy aggregate posterior (i.e. the Inception network identifies a wide variety of classes among the samples presented to it). Salimans et al. (2016) found this score correlated well with human evaluations of samplers trained on CIFAR-10; we therefore employ the Inception score here as a quantitative measure of visual fidelity of the samples, following the previous work's protocol of evaluating the average Inception score over 10 independent groups of 5,000 samples each. Error estimates correspond to standard deviations, in keeping with previously reported results.

## 3 Improving Unsupervised GAN Training On Diverse Datasets

In this work, we focus on the apparent difficulty of training GANs to produce "object-like" samples when trained on diverse collections of natural images. While Salimans et al. (2016) make progress on this problem by employing labeled data and training the discriminator, here we aim to make progress on the unsupervised case. Nevertheless, our methods would be readily applicable to supervised, semi-supervised or (with slight modifications) conditional setting.

We begin from the slightly subtle observation that in realistic manifestations of the GAN training procedure, the discriminator's (negative) gradient with respect to a sample points in a direction of (infinitesimal) local improvement with respect to the discriminator's estimate of the sample being data; it *does not* necessarily point in the direction of a draw from the data distribution. Indeed, the literature is replete with instances of gradient descent with respect to the input of a classification model, particularly wide-domain natural image classifiers, producing ghostly approximations to a particular class exemplar (Le et al., 2012; Erhan et al., 2009; Yosinski et al., 2015) when this procedure is carried out without additional guidance, to say nothing of the problems posed by adversarial examples (Szegedy et al., 2013; Goodfellow et al., 2014b) and fooling examples (Nguyen et al., 2015).

While the gradient of the loss function defined by the discriminator may be a source of information mostly relevant to very local improvements, the discriminator itself is a potentially valuable source of compact descriptors of the training data. Many authors have noted the remarkable versatility of high-level features learned by convolutional networks (Donahue et al., 2014; Yosinski et al., 2014) and the degree to which high-level semantics can be reconstructed from even the deepest layers of a network (Dosovitskiy & Brox, 2016). Although non-stationary, the distribution of the high-level activations of the discriminator when evaluated on data is ripe for exploitation as an additional source of knowledge about salient aspects of the data distribution.

We propose in this work to track this distribution with a denoising auto-encoder $r(\cdot)$ trained on the discriminator's hidden states when evaluated on training data. Alain & Bengio (2014) showed that a denoising auto-encoder trained on data from a distribution $q(\mathbf{h})$ estimates via $r(\mathbf{h}) - \mathbf{h}$ the gradient of the true log-density, $\frac{\partial \log q(\mathbf{h})}{\partial \mathbf{h}}$. Hence, if we train the denoising auto-encoder on the transformed training data $\mathbf{h} = \Phi(\mathbf{x})$ with $\mathbf{x} \sim \mathcal{D}$, then $r(\Phi(\mathbf{x}')) - \Phi(\mathbf{x}')$ with $\mathbf{x}' = G(\mathbf{z})$ indicates in which direction $\mathbf{x}'$ should be changed in order to make $\mathbf{h} = \Phi(\mathbf{x}')$ more like those features seen with the data. Minimizing $||r(\Phi(\mathbf{x}')) - \Phi(\mathbf{x}')||^2$ with respect to $\mathbf{x}'$ would thus push $\mathbf{x}'$ towards higher probability configurations according to the data distribution in the feature space $\Phi(\mathbf{x})$. We thus evaluate the discriminator features $\Phi(\mathbf{x})$, and the denoising auto-encoder, on samples from the generator, and treat the denoiser's output reconstruction as a fixed target for the generator. We refer to this procedure as *denoising feature matching*, and employ it as a learning signal for the generator in addition to the traditional GAN generator objective.

Formally, let $G$ be the generator parameterized by $\theta_G$, and $D = d \circ \Phi$ be our discriminator composing feature extractor $\Phi(\cdot) : \mathbb{R}^n \to \mathbb{R}^k$ and a classifier $d(\cdot) : \mathbb{R}^k \to [0, 1]$. Let $C(\cdot) : \mathbb{R}^k \to \mathbb{R}^k$ be a corruption function to be applied at the input of the denoising auto-encoder when it is trained to denoise. The parameters of the discriminator $D$, comprising the parameters of both $d$ and $\Phi$, is trained as in Goodfellow et al. (2014a), while the generator is trained according to

$$\underset{\theta_G}{\arg\min} \, \mathbb{E}_{\mathbf{z} \sim p(\mathbf{z})} \left[ \lambda_{\text{denoise}} \| \Phi(G(\mathbf{z})) - r(\Phi(G(\mathbf{z}))) \|^2 - \lambda_{\text{adv}} \log D(G(z)) \right] \tag{4}$$

where $r(G(\mathbf{z}))$ is treated as constant with respect to gradient computations. Simultaneously, the denoiser $r(\cdot)$ is trained according to the objective

$$\underset{\theta_r}{\arg\min} \, \mathbb{E}_{\mathbf{x} \sim \mathcal{D}} \| \Phi(\mathbf{x}) - r(C(\Phi(\mathbf{x}))) \|^2 \tag{5}$$

## 3.1 EFFECT OF $\Phi$

The theory surrounding denoising auto-encoders applies when estimating a denoising function from a data distribution $p(vecx)$. Here, we propose to estimate the denoising auto-encoder in the space of discriminator features, giving rise to a distribution $q(\Phi(\mathbf{x}))$. A natural question is what effect this has on the gradient being backpropagated. This is difficult to analyze in general, as for most choices the mapping $\Phi$ will not be invertible, though it is instructive to examine the invertible case. Assuming an invertible $\Phi : \mathbb{R}^n \to \mathbb{R}^n$, let $J = \frac{\partial \Phi(\mathbf{x})}{\partial \mathbf{x}}$ be the Jacobian of $\Phi$, and $q(\Phi(\mathbf{x})) = p(\mathbf{x})|J|$. By the inverse function theorem, $J$ is also invertible (and is in fact the Jacobian of the inverse $\Phi^{-1}$). Applying the chain rule and re-arranging terms, taking advantage of the invertibility of $J$, we arrive at a straightforward relationship between the score of $q$ and the score of $p$:

$$\frac{\partial \log q(\Phi(\mathbf{x}))}{\partial \Phi(\mathbf{x})} = \frac{\partial \log [p(\mathbf{x})|J|]}{\partial \Phi(\mathbf{x})} \tag{6}$$

$$= \frac{\partial \log p(\mathbf{x})}{\partial \Phi(\mathbf{x})} + \frac{\partial \log \left| \frac{\partial \Phi(\mathbf{x})}{\partial \mathbf{x}} \right|}{\partial \Phi(\mathbf{x})} \tag{7}$$

$$= \left( \frac{\partial \log p(\mathbf{x})}{\partial \mathbf{x}} + \frac{\partial \log |J|}{\partial \mathbf{x}} \right) J^{-1} \tag{8}$$

where

$$\frac{\partial \log |J|}{\partial x_k} = \text{Tr} \left( J^{-1} \frac{dJ}{dx_k} \right) \tag{9}$$

and $\frac{dJ}{dx_k}$ is a matrix of scalar derivatives of elements of J with respect to $x_k$. Thus, we see that the gradient backpropagated to the generator in an ideal setting is the gradient of the data distribution $p(\mathbf{x})$ along with an additive term which accounts for the changes in the rate of volume expansion/contraction in $\Phi$ locally around $\mathbf{x}$. In practice, $\Phi$ is not invertible, but the added benefit of the denoiser-targeted gradient appears to reduce underfitting to the modes of $p$ in the generator, irrespective of any distortions $\Phi$ may introduce.

## 4 RELATED WORK

Denoising feature matching was originally inspired by *feature matching* introduced by Salimans et al. (2016) as an alternative training criterion for GAN generators, namely (in our notation)

$$\underset{\theta_G}{\arg\min} \, \left| \| \mathbb{E}_{\mathbf{x} \sim \mathcal{D}} \left[ \Phi(\mathbf{x}) \right] - \mathbb{E}_{\mathbf{z} \sim p(z)} \left[ \Phi(G(\mathbf{z})) \right] \| \right|^2 \tag{10}$$

Feature matching is equivalent to linear maximum mean discrepancy (Gretton et al., 2006), employing linear first moment matching in the space of discriminator features $\Phi(\cdot)$ rather than the more familiar kernelized formulation. When performed on features in the penultimate layer, Salimans et al. (2016) found that the feature matching criterion was useful for the purpose of improving

results on semi-supervised classification, using classification of samples from the generator as a sophisticated form of data augmentation. Feature matching was, however, less successful at producing samples with high visual fidelity. This is somewhat unsurprising given that the criterion is insensitive to higher-order statistics of the respective feature distributions. Indeed, a degenerate $G$ which deterministically reproduces a single sample $\hat{\mathbf{m}}$ such that $\Phi(\hat{\mathbf{m}}) = \mathbb{E}_{\mathbf{x} \in \mathcal{D}} \Phi(\mathbf{x})$ trivially minimizes (10); in practice the joint training dynamics of $D$ and $G$ do not appear to yield such degenerate solutions.

Rather than aiming to merely reduce linear separability between data and samples in the feature space defined by $\Phi(\cdot)$, denoising feature matching selects a more probable (according to the feature distribution implied by the data, as captured by the denoiser) feature space target for each sample produced by $G$ and regresses $G$ towards it. While an early loss of entropy in $G$ could result in the generator locking on to one or a few attractors in the denoiser's energy landscape, we observe that this does not happen when used in conjunction with the traditional GAN objective, and in fact that the combination of the two objectives is notably robust to the collapses often observed in GAN training, even without taking additional measures to prevent them.

This work also draws inspiration from Alain & Bengio (2014), which showed that a suitably trained denoiser learns an operator which locally maps a sample towards regions of high probability under the data distribution. They further showed that a suitably trained[1] reconstruction function $r(\cdot)$ behaves such that

$$r(\mathbf{x}) - \mathbf{x} \propto \frac{\partial \log p(\mathbf{x})}{\partial \mathbf{x}} \tag{11}$$

That is, $r(\mathbf{x}) - \mathbf{x}$ estimates the score of the data generating distribution, up to a multiplicative constant. Our use of denoising auto-encoders necessarily departs from idealized conditions in that the denoiser is estimated online from an ever-changing distribution of features.

Several approaches to GAN-like models have cast the problem in terms of learning an energy function. Kim & Bengio (2016) extends GANs by modeling the data distribution simultaneously with an energy function parameterized by a deep neural network (playing the role of the discriminator) and the traditional generator, carrying out learning with a learning rule resembling that of the Boltzmann machine (Ackley et al., 1985), where the "negative phase" gradient is estimated from samples from the generator. The energy-based GAN formulation of Zhao et al. (2016) resembles our work in their use of an auto-encoder which is trained to faithfully reconstruct (in our case, a corrupted, function of) the training data. The energy-based GAN *replaces* the discriminator with an auto-encoder, which is trained to assign low energy ($L_2$ reconstruction error) to training data and higher energy to samples from $G$. To discourage generator collapses, a "pull-away term" penalizes the normalized dot product in a feature space defined by the auto-encoder's internal representation. In this work, we preserve the discriminator, trained in the usual discriminative fashion, and in fact preserve the traditional generator loss, instead augmenting it with a source of complementary information provided by targets obtained from the denoiser. The energy-based GAN can be viewed as training the generator to seek fixed points of the autoencoding function (i.e. by backpropagating through the decoder and encoder in order to decrease reconstruction error), whereas we treat the output of $r(\cdot)$ as constant with respect to the optimization as in Lee et al. (2015). That is to say, rather than using backpropagation to steer the dynamics of the autoencoder, we instead employ our denoising autoencoder to augment the gradient information obtained by ordinary backpropagation.

Closest to our own approach, concurrent work on model-based super-resolution by Sønderby et al. (2016) trains a denoising auto-encoder on high-resolution ground truth and evaluates it on synthesized super-resolution images, using the difference between the original synthesized image and the denoiser's output as an additional training signal for refining the output of the super-resolution network. Both Sønderby et al. (2016) and our own work are motivated by the results of Alain & Bengio (2014) discussed above. Aside from addressing a different application area, our denoiser is learned on-the-fly from a high-level feature representation which is itself learned.

---

[1] In the limit of infinite training data, with isotropic Gaussian noise of some standard deviation $\sigma$.

## 5 EXPERIMENTS

We evaluate denoising feature matching on learning synthesis models from three datasets of increasing diversity and size: CIFAR-10, STL-10, and ImageNet. Although several authors have described GAN-based image synthesis models operating at $128 \times 128$ (Salimans et al., 2016; Zhao et al., 2016) and $256 \times 256$ (Zhao et al., 2016) resolution, we carry out our investigations at relatively low resolutions, both for computational ease and because we believe that the problem of unconditional modeling of diverse image collections is not well solved even at low resolutions; making progress in this regime is likely to yield insights that apply to the higher-resolution case.

In all experiments, we employ isotropic Gaussian corruption noise with $\sigma = 1$. Although we experimented with annealing $\sigma$ towards 0 (as also performed in Sønderby et al. (2016)), an annealing schedule which consistently outperformed fixed noise remained elusive. We experimented with convolutional denoisers, but our best results to date were obtained with deep, fully-connected denoisers using the ReLU nonlinearity on the penultimate layer of the discriminator. The number of hidden units was fixed to the same value in all denoiser layers, and the procedure is apparently robust to this hyperparameter choice, as long as it is greater than or equal to the input dimensionality.

Our generator and discriminator architectures follow the methods outlined in Radford et al. (2015). Accordingly, batch normalization (Ioffe & Szegedy, 2015) was used in the generator and discriminator in the same manner as Radford et al. (2015), and in all layers of the denoiser except the output layer. In particular, as in Radford et al. (2015), we separately batch normalize data and generator samples for the discriminator and denoiser with respect to each source's statistics. We calculate updates with respect to all losses with the parameters of all three networks fixed, and update all parameters simultaneously.

All networks were trained with the Adam optimizer Kingma & Ba (2014) with a learning rate of $10^{-4}$ and $\beta_1 = 0.5$. The Adam optimizer is scale invariant, and so it suffices to e.g. tune $\lambda_{\text{denoise}}$ and fix $\lambda_{\text{adv}}$ to 1. In our experiments, we set $\lambda_{denoise}$ to $0.03/n_h$, where $n_h$ is the number of discriminator hidden units fed as input to the denoiser; this division decouples the scale of the first term of (4) from the dimensionality of the representation used, reducing the need to adjust this hyperparameter simply because we altered the architecture of the discriminator.

### 5.1 CIFAR-10

CIFAR-10 (Krizhevsky & Hinton, 2009) is a small, well-studied dataset consisting of 50,000 $32 \times 32$ pixel RGB training images and 10,000 test images from 10 classes: airplane, automobile, bird, cat, deer, dog, frog, horse, ship, and truck.

Samples from our model trained on CIFAR-10 are shown in Figure 1, and Inception scores for several methods, including those reported in Salimans et al. (2016) and scores computed from samples generated from a model presented in Dumoulin et al. (2016), are presented in Table 1. We achieve a mean Inception score of 7.72, falling slightly short of Salimans et al. (2016), which employed a supervised discriminator network (the same work reports a score of $4.36 \pm .04$ when labels are omitted from their training procedure). Qualitatively, the samples include recognizable cars, boats and various animals. The best performing generator network consisted of the $32 \times 32$ ImageNet architecture from Radford et al. (2015) with half the number of parameters at each layer, and less than 40% of the parameters of the CIFAR-10 generator presented in Salimans et al. (2016).

| Real data[*] | Semi-supervised | | Unsupervised | |
| --- | --- | --- | --- | --- |
| | Improved GAN (Salimans *et al*)[*] | | ALI (Dumoulin *et al*)[†] | Ours |
| $11.24 \pm .12$ | $8.09 \pm .07$ | | $5.34 \pm 0.05$ | $7.72 \pm 0.13$ |

Table 1: Inception scores for models of CIFAR-10. [*] as reported in Salimans et al. (2016); semi-supervised [†] computed from samples drawn using author-provided model parameters and implementation.

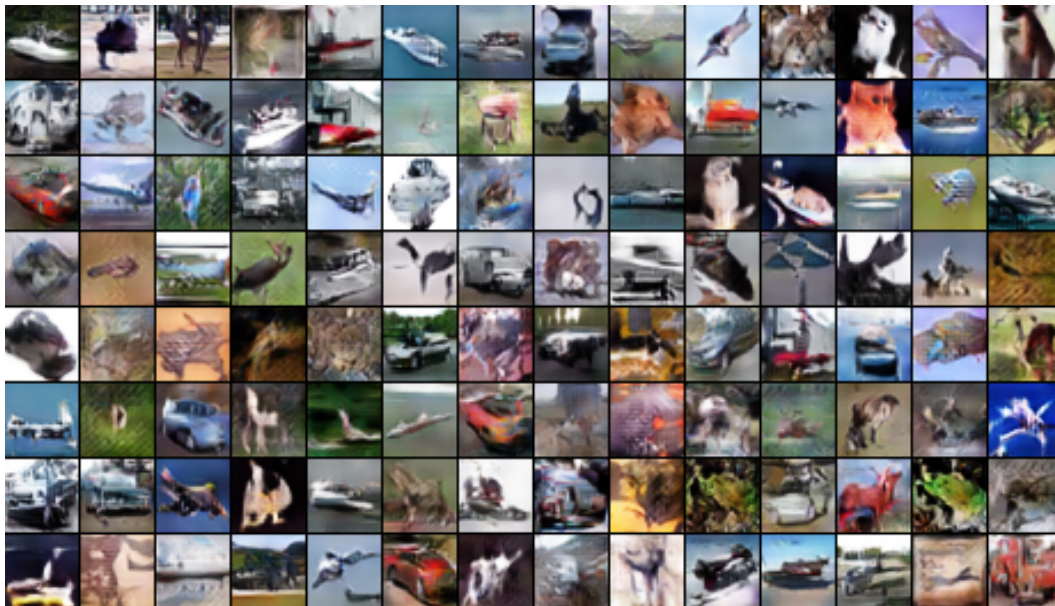

Figure 1: Samples generated from a model trained with denoising feature matching on CIFAR10.

## 5.2 STL-10

STL-10 (Coates et al., 2011) is a dataset consisting of a small labeled set and larger (100,000) unlabeled set of $96 \times 96$ RGB images. The unlabeled set is a subset of ImageNet that is more diverse than CIFAR-10 (or the labeled set of STL-10), but less diverse than full ImageNet. We downsample by a factor of 2 on each dimension and train our networks at $48 \times 48$. Inception scores for our model and a baseline, consisting of the same architecture trained without denoising feature matching (both trained for 50 epochs), are shown in Table 2. Samples are displayed in Figure 2.

| Real data | Ours | GAN Baseline |
|---|---|---|
| $26.08 \pm .26$ | $8.51 \pm 0.13$ | $7.84 \pm .07$ |

Table 2: Inception scores for models of the unlabeled set of STL-10.

## 5.3 IMAGENET

The ImageNet database (Russakovsky et al., 2014) is a large-scale database of natural images. We train on the designated training set of the most widely used release, the 2012 ImageNet Large Scale Visual Recognition Challenge (ILSVRC2012), consisting of a highly unbalanced split among 1,000 object classes. We preprocess the dataset as rescaled central crops following the procedure of Krizhevsky et al. (2012), except at $32 \times 32$ resolution to facilitate comparison with Radford et al. (2015).

ImageNet poses a particular challenge for unsupervised GANs due to its high level of diversity and class skew. With a generator and discriminator architecture identical to that used for the same dataset in Radford et al. (2015), we achieve a higher Inception score using denoising feature matching, using denoiser with 10 hidden layers of 2,048 rectified linear units each. Both fall far short of the score assigned to real data at this resolution; there is still plenty of room for improvement. Samples are displayed in Figure 3.

## 6 DISCUSSION AND FUTURE DIRECTIONS

We have shown that training a denoising model on high-level discriminator activations in a GAN, and using the denoiser to propose high-level feature targets for the generator, can usefully improve

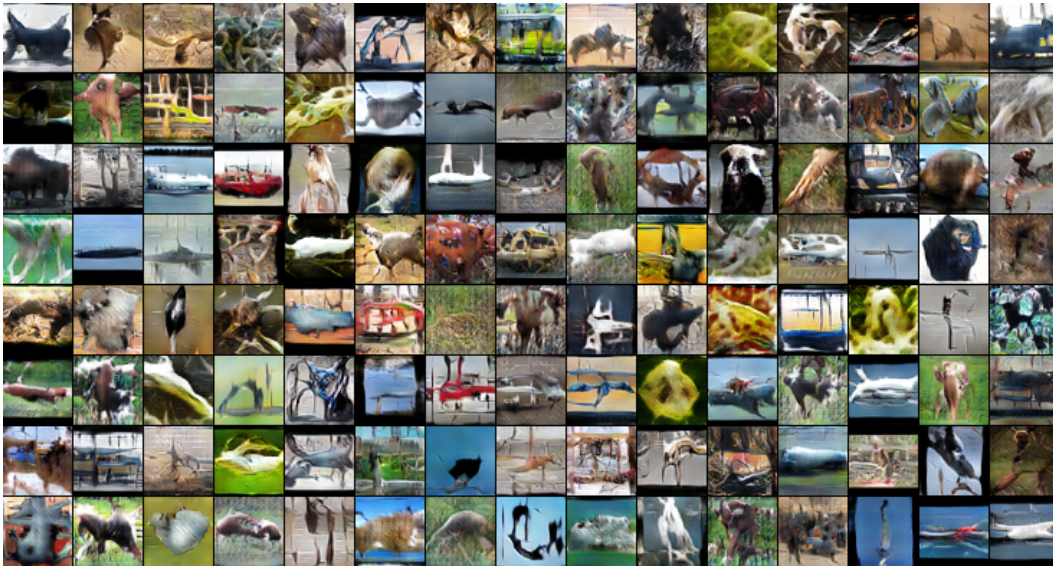

Figure 2: Samples from a model trained with denoising feature matching on the unlabeled portion of the STL-10 dataset.

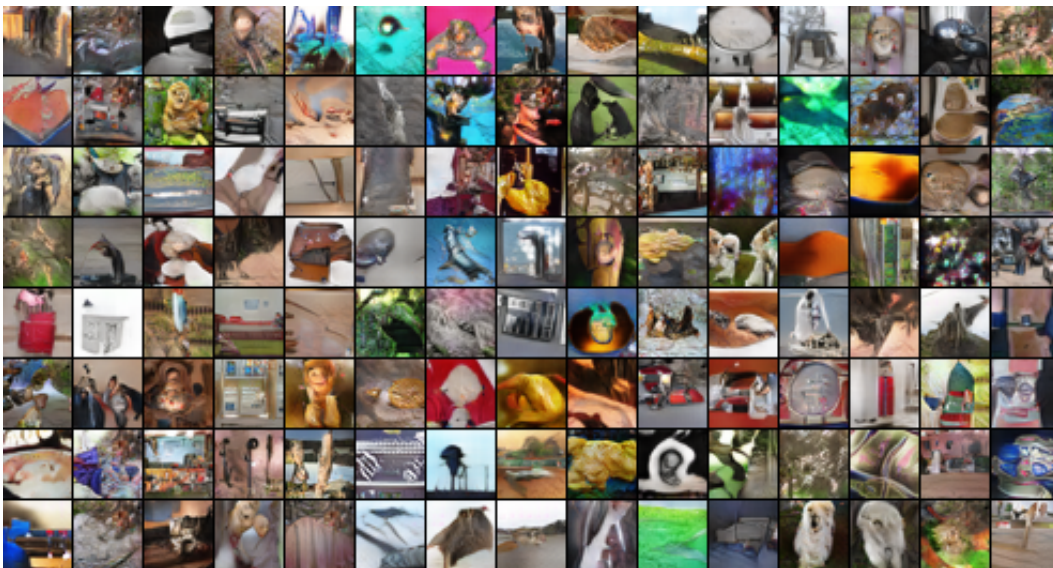

Figure 3: Samples from our model of ILSVRC2012 at $32 \times 32$ resolution.

| Real data | Radford *et al*[⋆] | Ours |
|---|---|---|
| $25.78 \pm .47$ | $8.83 \pm 0.14$ | $9.18 \pm .13$ |

Table 3: Inception scores for models of ILSVRC 2012 at $32 \times 32$ resolution. [⋆] computed from samples drawn using author-provided model parameters and implementation.

GAN image models. Higher Inception scores, as well as visual inspection, suggest that the procedure captures class-specific features of the training data in a manner superior to the original adversarial objective alone. That being said, we do not believe we are yet making optimal use of the paradigm. The non-stationarity of the feature distribution on which the denoiser is trained could be limiting the ability of the denoiser to obtain a good fit, and the information backpropagated to the generator is always slightly stale. Steps to reduce this non-stationarity may be fruitful; we experimented briefly with historical averaging as explored in Salimans et al. (2016) but did not observe a clear benefit thus far. Structured denoisers, including denoisers that learn an energy function for multiple hidden layers at once, could conceivably aid in obtaining a better fit. Learning a partially stochastic transition operator rather than a deterministic denoiser could conceivably capture interesting multimodalities that are "blurred" by a unimodal denoising function.

Our method is orthogonal and could conceivably be used in combination with several other GAN extensions. For example, methods incorporating an encoder component (Donahue et al., 2016; Dumoulin et al., 2016), various existing conditional architectures (Mirza & Osindero, 2014; Denton et al., 2015; Reed et al., 2016), or the semi-supervised variant employed in Salimans et al. (2016), could all be trained with an additional denoising feature matching objective.

We have proposed a useful heuristic, but a better theoretical grounding regarding how GANs are trained in practice is a necessary direction for future work, including grounded criteria for assessing mode coverage and mass misassignment, and principled criteria for assessing convergence or performing early stopping.

### ACKNOWLEDGMENTS

We thank Ian Goodfellow, Laurent Dinh, Yaroslav Ganin and Kyle Kastner for helpful discussions. We thank Vincent Dumoulin and Ishmael Belghazi for making available code and model parameters used in comparison to ALI, as well as Alec Radford for making available the code and model parameters for his ImageNet model. We would like to thank Antonia Creswell and Hiroyuki Yamazaki for pointing out an error in the initial version of this manuscript, and anonymous reviewers for valuable feedback. We thank the University of Montreal and Compute Canada for the computational resources used for this investigation, as well as the authors of Theano (Al-Rfou et al., 2016), Blocks and Fuel (van Merriënboer et al., 2015). We thank CIFAR, NSERC, Google, Samsung and Canada Research Chairs for funding.

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
