# Peer review of "Improving Generative Adversarial Networks with Denoising Feature Matching"

_ICLR 2017 — accepted_

[Public Comment · Antonia Creswell · 16 Nov 2016]
**Training Scheme and Denoising**

The generations in this paper suggest that using extra information from features of the discriminator allows the generator to produce images with more object like features. I have some questions/comments:

1) In equation 5 it appears that you are training r to reconstruct a corrupted version of the features, rather than the features themselves, the reason for this is not clear?
|| C(phi(x)) - r(C(phi(x))) ||  rather than  || phi(x) - r(C(phi(x))) ||

2) This approach involves training 3 networks. It would be interesting to know what kind of training scheme was used? Whether, D,G and r networks are trained for one iteration each, or if some networks are trained for more iterations before updating the next network? 

3) It would also be interesting to know whether parameters l_denoise and l_adv are fixed or adjusted during training?

[Official Review · AnonReviewer3 · rating 6 · confidence 2 · 16 Dec 2016]
**GANtastic paper**
soundness 3

This paper is about using denoising autoencoders to improve performance in GANs. In particular, the features as determined by the discriminator, of images generated by the generator, are fed into a denoising AE and we try to have these be reconstructed well. I think it's an interesting idea to use this "extra information" -- namely the feature representations learned by the discriminator. It seems very much in the spirit of ICLR! My main concern, though, is that I'm not wholly convinced on the nature of the improvement. This method achieves higher inception scores than other methods in some cases, but I have a hard time interpreting these scores and thus a hard time getting excited by the results. In particular, the authors have not convinced me that the benefits outweigh the required additional sophistication both conceptually and implementation-wise (speaking of which, will code be released?). One thing I'd be curious to know is, how hard is it to get this thing to actually work? 

Also, I view GANs as a means to an end -- while I'm not particularly excited about generating realistic images (especially in 32x32), I'm very excited about the future potential of GAN-based systems. So it would have been nice to see these improvements in inception score translate into improvements in a more useful task. But this criticism could probably apply to many GAN papers and so perhaps isn't fair here. 

I do think the idea of exploiting "extra information" (like discriminator features) is interesting both inside and outside the context of this paper.

[Official Review · AnonReviewer2 · rating 7 · confidence 5 · 16 Dec 2016]
soundness 5 · clarity 3

The authors present a way to complement the Gerative Adversarial Network traning procedure with an additional term based on denoising autoencoders. The use of denoising autoencoders is motivated by the observation that they implicitly capture the distribution of the data they were trained on. While sampling methods based denoising autoencoders alone don't amount to very interesting generative models (at least no-one could demonstrate otherwise), this paper shows that it can be combined successfully with generative adversarial networks.

My overall assessment of this paper is that it is well written, well reasoned, and presents a good idea motivated from first principles. I feel that the idea presented here is not revolutionary or a very radical departure from what has been done before, I would have liked a slightly more structured experiments section which focusses on and provides insights into the relative merits of different choices one could make (see pre-review questions for details), rather than focussing just on demonstrating that a chosen variant works.

In addition to this general review, I have already posted specific questions and criticism in the pre-review questions - thanks for the authors' responses. Based on those responses the area I am most uncomfortable about is whether the (Alain & Bengio, 2014) intuition about the denoising autoencoders is valid if it all happens in a nonlinear featurespace. If the denoiser function's behaviour ends up depending on the Jacobian of the nonlinear transformation Phi, another question is whether this dependence is exploitable by the optimization scheme.

[Official Review · AnonReviewer1 · rating 7 · confidence 4 · 18 Dec 2016]
**No Title**
originality 4 · clarity 5

This paper is well written, and well presented. This method is using denoise autoencoder to learn an implicit probability distribution helps reduce training difficulty, which is neat. In my view, joint training with an auto-encoder is providing extra auxiliary gradient information to improve generator. Providing auxiliary information may be a methodology to improve GAN.  
 
Extra comment:
Please add more discussion with EBGAN in next version.

[Author Response · David Warde-Farley · 19 Jan 2017]
**Revision**

A revision has been posted, addressing several textual concerns, correcting an error in equation 5, and including discussion of the effect of the feature space Phi on the score function learned by the DAE. A further update will follow later this week if any of the additional experiments bear fruit. Apologies again for the delay.

[Final Decision · Program Chairs · 06 Feb 2017]
**ICLR committee final decision**

The idea of using a denoising autoencoder on features of the discriminator is sensible, and explored and described well here. The qualitative results are pretty good, but it would be nice to try some of the more recent likelihood-based methods for quantitative evaluation, as the inception score is not very satisfying. Also it would be interesting to see if this additional term helps in scaling up to larger images.